# Anticoagulants utilization in eight hospitals within the Luzhou region from 2019 to 2023

Wei Luo[1,2☯], Yan Li[3☯], Jiali Yang[1,2☯], Yang Liu[1,2], Yue Shi[1,2], Hongli Luo[1]*

1 Department of Pharmacy, The Affiliated Hospital of Southwest Medical University, Luzhou, China,
2 School of Pharmacy, Southwest Medical University, Luzhou, China, 3 Department of Pharmacy, The First Afflicted Hospital of Chengdu Medical College, Chengdu, China

☯ These authors contributed equally to this work.
* lyfylhl@163.com

**Data Availability Statement:** The data supporting the results of this study were obtained from the hospital information system and the automatic prescription screening system of Sichuan Meke Software Research and Development Co., LTD. The

## Abstract

### Background

With the increasing utilization of anticoagulants, the selection of appropriate anticoagulants has emerged as a significant quandary. The objective of this study was to evaluate recent trend in the utilization and expenditure of anticoagulants within a specific region, aiming to provide valuable insights into the optimal choice of anticoagulants across other healthcare facilities.

### Methods

The utilization of anticoagulants was retrospectively analyzed. The data on anticoagulant utilizations in tertiary-care hospitals within a district were collected from January 2019 to December 2023. The expenditure, defined daily doses (DDDs), and defined daily cost (DDC) were calculated. The trends in the utilization and expenditure of anticoagulants were examined using linear regression analysis.

### Results

From 2019 to 2023, the DDDs of rivaroxaban demonstrated a significant annual increase in most hospitals ($p < 0.05$). Only a few hospitals exhibited a gradual rise in the consumption of low molecular weight heparin (LMWH) over the same period ($p < 0.05$). The trend of heparin sodium and warfarin varied across different hospitals. The implementation of the centralized procurement policy, however, resulted in a decline in the consumption of rivaroxaban and LMWH in 2021 and 2022 respectively. The DDC value of rivaroxaban experienced a substantial decrease over the past five years ($p = 0.020$), declining from 55.20 Chinese Yuan (CNY) in 2019 to 4.28 CNY in 2023. Conversely, there was a slight increase noted in the DDC of heparin sodium during this time frame ($p = 0.042$).

### Conclusion

Over the past five years (2019–2023), there has been an increase in the utilization of rivaroxaban and LMWH. However, their expenditure has decreased. In addition, the utilization

availability of these data is limited, and all data of this study are provided in the Supplementary Information.

**Funding:** Sichuan Provincial Department of Science and Technology project (No. 2023JDKP0040). The funders had a role in Conceptualization, Methodology, preparation of the manuscript, and decision to publish.

**Competing interests:** The authors declare there are no conflicts of interest.

and expenditure of warfarin and heparin sodium remained relatively stable. The application prospects of rivaroxaban and LMWH are promising.

## Introduce

Venous thromboembolism (VTE) is a prevalent cardiovascular disorder characterized by the formation of blood clots in the lower extremity veins, which subsequently dislodge and migrate to the heart and lungs. This condition encompasses two main entities: deep vein thrombosis (DVT) and pulmonary embolism (PE) [1]. VTE affects millions of individuals globally each year and is associated with morbidity and mortality rates comparable to those observed in myocardial infarction. [2]. In Europe and the United States, the estimated VTE incidence ranged from 1 per 1000 to 2 per 1000 population. Comparatively, Asia exhibits a lower rate of VTE compared to Europe and the United States [3]. Notably in China, VTE prevention has been listed as one of the top ten national goals for medical safety improvement for four consecutive years (2021–2024) [4–7]. The prevention of VTE is feasible. However, the presence of obesity, diabetes, smoking, hypertension and hyperlipidemia can elevate the risk of VTE [8, 9]. Approximately 30% of patients diagnosed with VTE experience recurrence within a decade [10]. Furthermore, VTE not only poses a risk of mortality in the first year following diagnosis but also remains an important cause of death during the subsequent 30-year follow-up period. Mortality from VTE and other cardiovascular, cancer, and respiratory diseases is higher in patients with VTE compared with the general population [11].

Anticoagulant therapy constitutes the cornerstone in the management of VTE. The progression of VTE is typically categorized into three distinct stages during the course of the disease: an initial acute phase lasting for 5–10 days following the diagnosis of VTE, a subsequent maintenance phase spanning 3–6 months, and finally an extended phase extending beyond this designated timeframe. Diverse stages entail distinct demands for the utilization of anticoagulants [8, 12]. Anticoagulants can be broadly classified into three categories: vitamin K antagonists (VKA), direct oral anticoagulants (DOACs), and parenteral anticoagulants. Clinical indications for each anticoagulants should be considered based on the risk of adverse drug events, Pharmacokinetics and therapeutic drug monitoring [13, 14].

The use of anticoagulants when the main adverse reaction is increased risk of bleeding [15]. In a meta-analysis of 33 studies involving 4374 patients receiving oral anticoagulant therapy, the case fatality rate for major bleeding was determined to be 13.4% (95% CI, 9.4% to 17.4%) [16]. VKA drugs, exemplified by warfarin, exhibit a narrow therapeutic index necessitating regular monitoring of patient prothrombin time to maintain an international normalized ratio (INR) within the range of 2.0 to 3.0 [17]. Over a span of five years, a prospective study on warfarin demonstrated that by adjusting the treatment strategy based on INR, the incidence rate of adverse drug reactions (ADRs) associated with warfarin significantly decreased from 3.8% to 0.98% ($p < 0.0001$) [18]. The primary guideline recommendation is to prioritize DOACs for the treatment of VTE, as opposed to the vitamin K antagonist, due to DOACs exemption from regular INR monitoring [19]. Furthermore, DOACs exhibit a low risk of bleeding and drug interactions, while also demonstrating minimal impact on food-drug interactions [8, 20].

With the publication of "Good Manufacturing Practice for Drugs", medical institutions have increasingly prioritized drug quality control, encompassing not only pharmaceutical production, drug procurement, and storage but also rational clinical drug utilization [21]. By fortifying quality supervision, the standard of rational drug utilization is enhanced. The field of drug utilization research (DUR) employs a range of descriptive and analytical methods to

quantitatively assess, comprehend, and evaluate the processes involved in drug prescribing, dispensing, and consumption [22]. DUR can be utilized to identify early indicators of inappropriate drug utilization, thereby promoting the safe and effective utilization of medications and enhancing the quality of drug therapy [23].

The city of Luzhou, situated in the southern region of Sichuan province, stands as a representative example with relatively high medical and economic standards within the province. Moreover, owing to factors such as advanced medical facilities and convenient transportation, tertiary-care hospitals are predominantly chosen by patients for their treatment needs. Consequently, this inclination towards tertiary-care hospitals results in a greater utilization of anticoagulant medications and facilitates a more comprehensive understanding of their usage trends. Therefore, the objective of this study is to provide an all-encompassing overview regarding the utilization and expenditure patterns of anticoagulant drugs in Luzhou's tertiary-care hospitals from 2019 to 2023 through DUR.

## Methods

### Data source

This study was a retrospective multicenter study in Luzhou City, and this study's observation period spanned from 2019 to 2023. The 8 hospitals were categorized into Grade A (A, B) and Grade B (C, D, E, F, G, H) based on the implementation rules of tertiary-care hospital accreditation standards [24]. Data pertaining to anticoagulants between January 2019 and December 2023 were extracted from the hospital information system (HIS).

### Data collection

The extracted data includes prescription drugs' general and commodity names, strength, amounts, and expenditure (S2 and S3 Tables). The analysis encompassed VKA (Warfarin), DOACs (rivaroxaban, apixaban, edoxaban, and dabigatran etexilate), and parenteral anticoagulants (heparin sodium, heparin calcium, enoxaparin, nadroparin, dalteparin, fondaparinux, argatroban, and bivalirudin). Information on individual participants could not be accessed and identified during and after data collection.

The present study conducted a retrospective analysis on the utilization of anticoagulant medications in hospitals located in Luzhou, covering the period from January 2019 to December 2023. Types of drugs, expenditure, defined daily dose (DDD), such as statistics, and compare the use present situation and trend of anticoagulants. Among them, heparins with lower molecular weight such as enoxaparin, nadroparin and dalteparin are uniformly referred to as low molecular weight heparin (LMWH).

The expenditure was determined using the Chinese yuan (CNY) to determine the expenditure of anticoagulants, and 7.27 CNY was equal to 1 US dollar. DDD is the assumed average maintenance dose per day of a drug used for its main indication in adults [25]. Defined daily doses (DDDs) were calculated according to DDD values provided by WHO (For example, the DDD value for warfarin is 7.5mg) [26]. DDDs = annual consumption of a drug/DDD value of the drug. The DDDs are primarily employed for drug utilization assessment. A higher value of DDDs indicates a greater frequency of drug use, thereby reflecting a clinical inclination towards the choice of the drug. The defined daily cost (DDC) was utilized as a measure to accurately depict the financial implications and economic burden associated with each medication [27]. DDC = an annual consumption amount of the drug/drug DDDs value. A higher DDC value indicates a more substantial economic burden on patients.

The selection of warfarin as a prescription was made to assess the rationality of its usage. The warfarin leaflet and "China expert consensus of warfarin anticoagulant therapy" [28],

"Guidelines on oral anticoagulation with warfarin–fourth edition" [29] serve as the standard for evaluating indications of warfarin use.

## Statistical analysis

The statistical indicators of eight hospitals in Luzhou were calculated using Excel software. For substances with different specifications and manufacturers, the quantities were standardized through unit conversion or summation to obtain the total amount of the drug. Subsequently, Graphpad Prism 8.0 software was used to produce graphs. The trends in anticoagulants use were analyzed using linear regression analysis in SPSS 24.0, and $p$ values less than 0.05 were considered statistically significant.

## Ethics statement

The data used in this study were completely anonymized, confirmed by the ethics committee, did not require review and approval by the ethics committee, and the requirement for informed consent was waived.

## Results

### Utilization and trends of anticoagulants

The current landscape of anticoagulants encompasses various categories, including parenteral anticoagulants, VKA, and DOACs. In recent years (2019–2023), the utilization of anticoagulants in Luzhou eight hospitals reveals (Fig 1) that the predominant anticoagulants employed are warfarin, rivaroxaban, heparin sodium, and LMWH. For Hospital A, the top four DDDs were LMWH (1605935), warfarin (1236290), rivaroxaban (911944), and heparin sodium (86410). While other anticoagulants are being used, it can be observed that Third-Level Grade A hospitals prescribing various medications exceeds that of Third-Level Grade B hospitals. Considering the different types of drugs used in different hospitals, four anticoagulants, warfarin, rivaroxaban, heparin sodium and LMWH, which were used in eight hospitals were selected as the subjects of follow-up study.

From 2019 to 2023, except for hospitals F and G, there was a significant year-on-year increase in DDDs of rivaroxaban in the other six hospitals ($p < 0.05$) (S1 Table, Fig 2B). It can be observed that tertiary-care class B hospitals had minimal usage of rivaroxaban during 2019–2020. However, substantial use of rivaroxaban in these hospitals occurred only from 2021 to

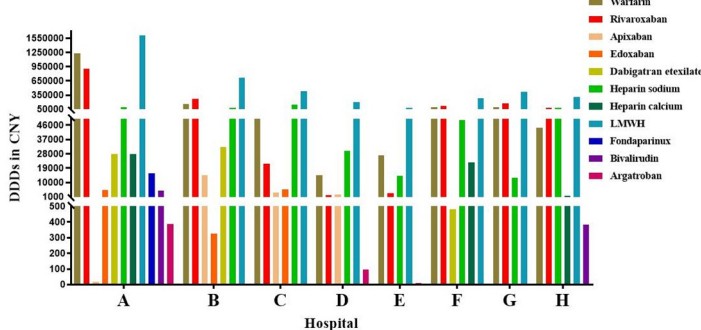

**Fig 1. Utilization of anticoagulants at eight hospitals from 2019 to 2023.** DDDs: defined daily doses; LMWH: low molecular weight heparin.

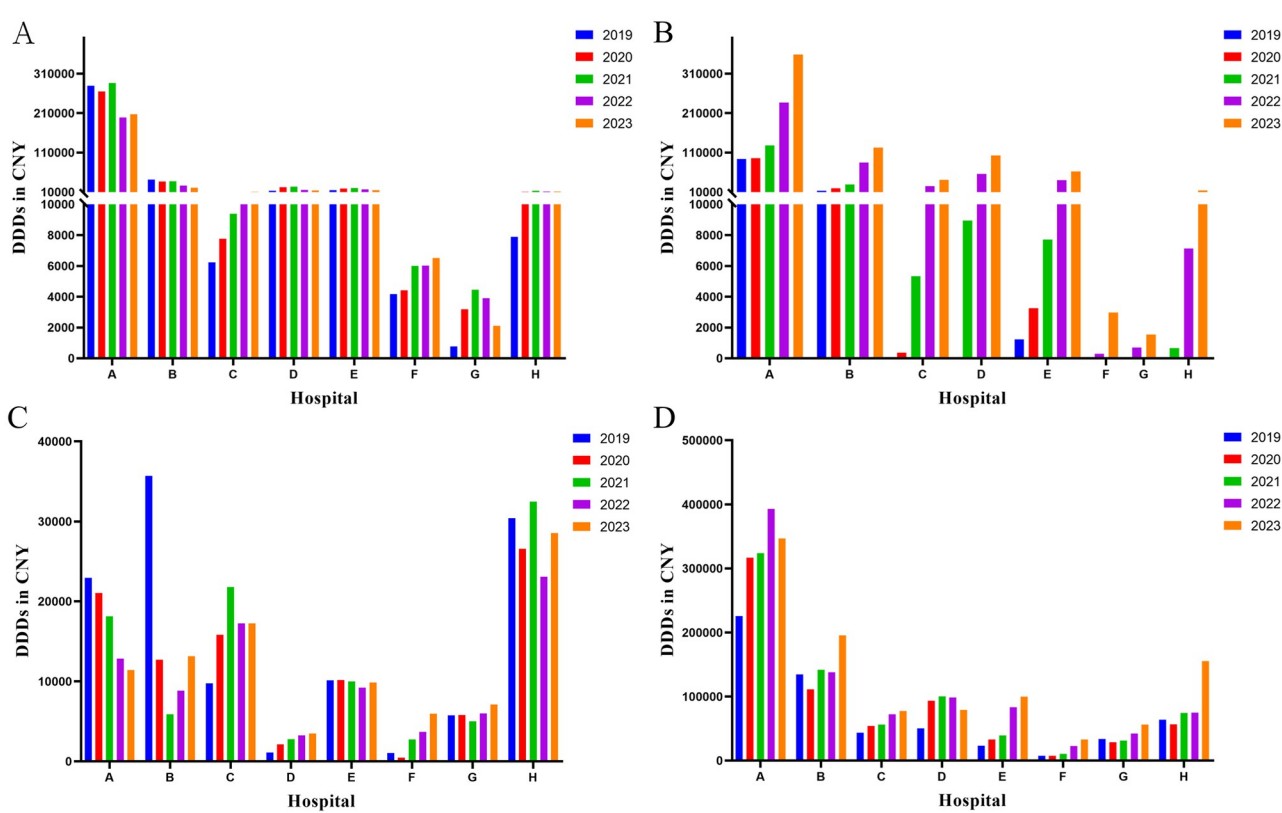

**Fig 2. Utilization of anticoagulants from 2019 to 2023.** A: warfarin. B: rivaroxaban; C: heparin sodium; D: LMWH. DDDs: defined daily doses; LMWH: low molecular weight heparin.

2023. For instance, Hospital H did not employ rivaroxaban in both 2019 and 2020; however, its DDDs increased from 662 in 2021 to 13,795 in 2023 after initiation of usage. Furthermore, although there was a significant year-on-year increase ($p < 0.05$) observed only in hospitals C, E, and F for LMWH DDDs from 2019 to2023(S1 Table, Fig 2D), the overall use of LMWH remained higher than that of heparin sodium across all hospitals within the same year. Conversely, heparin sodium and warfarin demonstrated variable trends among different hospitals over the past five years (S1 Table, Fig 2A and 2C).

## Expenditure of anticoagulants utilization

The comparison of anticoagulants usage and expenditure amounts reveal a similar trend in the sum of warfarin and heparin sodium expenditure, as well as changes in DDDs (Fig 3A and 3C). Notably, the expenditure of rivaroxaban and LMWH exhibited divergent trends in terms of DDDs. For rivaroxaban, the utilization of DDDs increased across the eight hospitals from 2019 to 2023; however, there was a decrease in expenditure observed from 2021 to 2023 (Fig 3B). Similarly, the expenditure of LMWH exhibit a decreasing trend in 2022–2023 (Fig 3D). In A hospital, the DDDs for rivaroxaban increased from 128,232 in 2021 to 358,435 in 2023. Additionally, the corresponding expenditure decreased from 5.8244 million CNY in 2021 to 2.2955 million CNY in 2023.

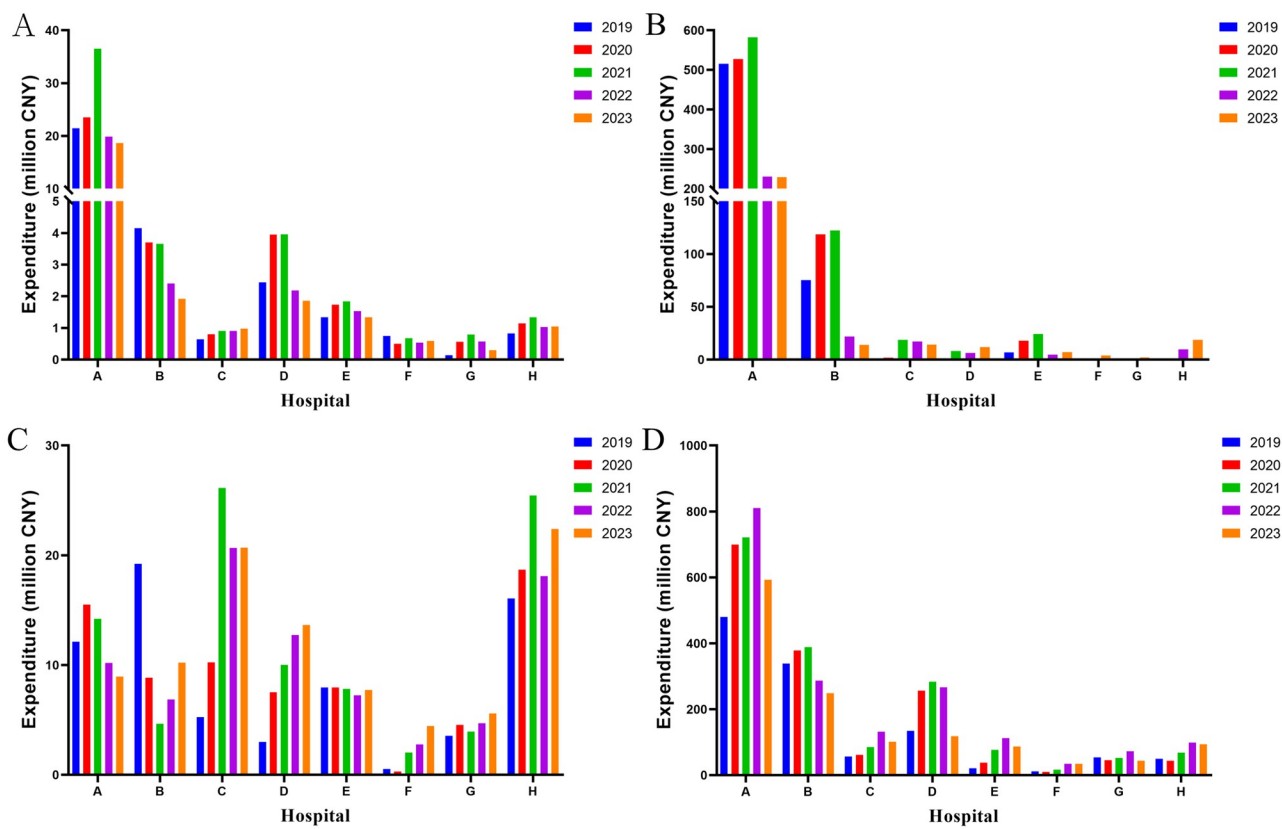

**Fig 3. The expenditure on anticoagulants in the period of 2019-2023(million CNY).** A: Warfarin; B: Rivaroxaban; C: Heparin sodium; D: LMWH. LMWH: low molecular weight heparin.

## Outpatient vs inpatient utilization of anticoagulants

In all eight hospitals, warfarin, rivaroxaban, and LMWH were the predominant anticoagulants administered in the outpatient department, with their utilization surpassing that of heparin sodium (Fig 4). Despite the high dosage of LMWH in terms of DDDs, there were still discernible distinctions compared to the overall quantity. Taking Hospital A as an example, the outpatient DDDs of warfarin, rivaroxaban, and LMWH in 2023 were recorded as 185,580, 237,453, and 46,401 respectively, which accounting for 89.6%, 66.2% and 13.4% of the total DDDs in the hospital.

The administration of LMWH in inpatient surpasses that of the other three drugs by a margin (Fig 5). Moreover, a substantial escalation in the utilization of heparin sodium was observed within the hospital setting when compared to DDDs of outpatient heparin sodium. Taking Hospital A as an example, the hospitalization DDDs of heparin sodium and LMWH in 2023 were 11,269 and 300,236, respectively, accounting for 98.6% and 86.6% of the total DDDs administered at the hospital. Furthermore, the findings indicate a prevailing utilization of parenteral anticoagulants in hospital settings for the administration of anticoagulants. Rivaroxaban also holds an important position among inpatient anticoagulant medications in hospital A, accounting for 33.8% of the total in 2023.

Due to the substantial volume of prescriptions in both outpatient and inpatient settings, the rational utilization of medications has emerged as an important concern. The prescription of

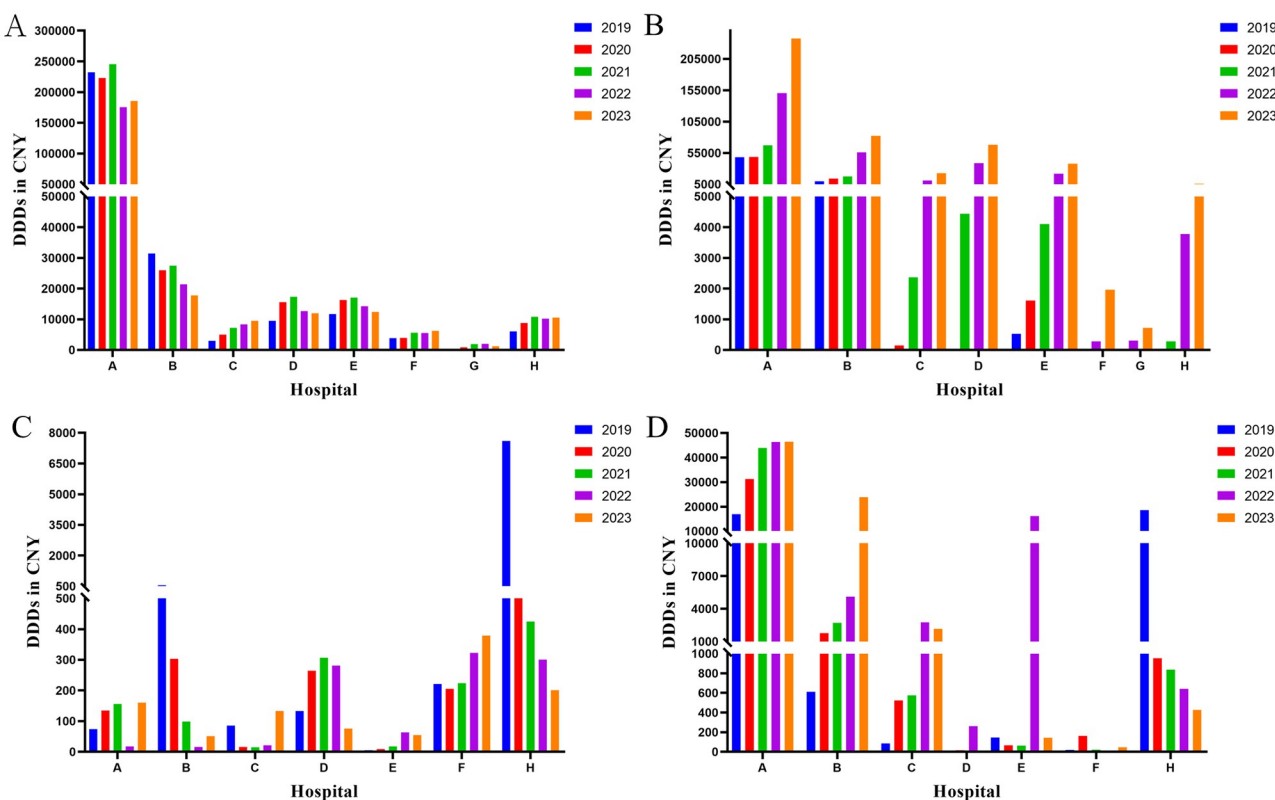

**Fig 4. Outpatient utilization of anticoagulants from 2019 to 2023.** A: Warfarin; B: Rivaroxaban; C: Heparin sodium; D: LMWH. DDDs: defined daily doses; LMWH: low molecular weight heparin.

warfarin in hospital A in 2023 was used as an exemplification to comprehend the indications for warfarin administration. In 2023, the retrieval of warfarin prescriptions was categorized into inpatient (n = 734) and outpatient (n = 6224) prescriptions. According to the prescription diagnosis, the classification includes mechanical valve replacement status, atrial fibrillation, venous thrombosis, cerebral infarction, cardiac surgery (valvular surgery, implantable cardioverter defibrillator (ICD), Bentall), heart valve disease and other indications (Table 1). The utilization of warfarin in the outpatient setting predominantly pertained to mechanical valve replacement (74.1%), followed by atrial fibrillation (20.6%). However, the utilization of warfarin in the inpatient setting predominantly pertained to cardiac valvular disease (54.4%), followed by atrial fibrillation (46.0%). Due to a prescription may contain a variety of indications, so these prescriptions will be superposition calculation.

## DDC and trends of anticoagulants

The DDC represents the average daily cost value, and a higher DDC indicates increased economic burden on patients. DDC value is the lowest drug warfarin, nearly five years are about 1 CNY, belong to the burden lighter (Fig 6). And the DDC value of rivaroxaban, heparin sodium, and LMWH is relatively high. The DDC value of rivaroxaban exhibited a significant decrease over the past 5 years ($p = 0.020$), declining from 55.20 CNY in 2019 to 4.28 CNY in 2023. Furthermore, there was a slight increase observed in the DDC of heparin sodium during the same period ($p = 0.042$), rising from 5.79 CNY in 2019 to 9.67 CNY in 2023. However, the

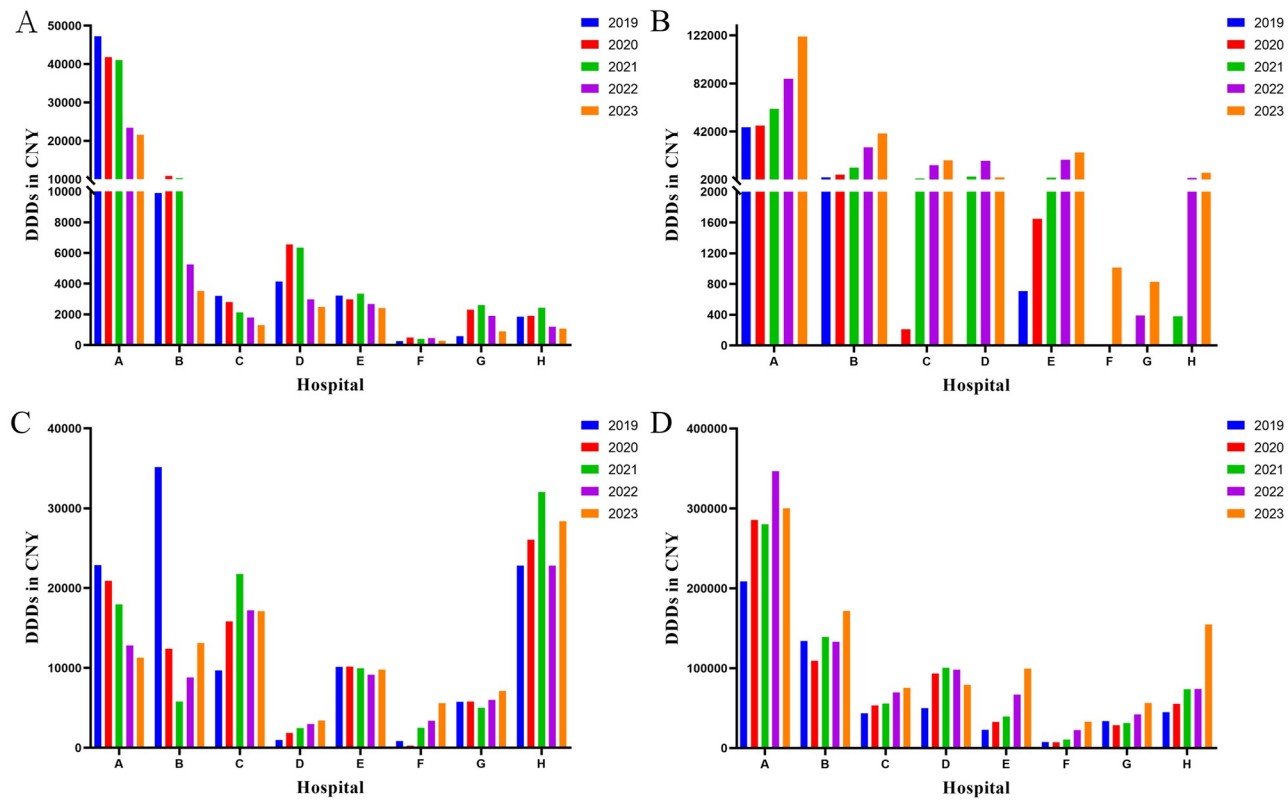

**Fig 5. Inpatient utilization of anticoagulants from 2019 to 2023.** A: Warfarin; B: rivaroxaban; C: Heparin sodium; D: LMWH. DDDs: defined daily doses; LMWH: low molecular weight heparin.

DDC values of warfarin ($p = 0.793$) and LMWH ($p = 0.202$) remained relatively stable throughout the past 5 years.

## Discussion

The findings of this study present the most up-to-date data on the utilization and expenditure patterns of anticoagulants in tertiary-care hospitals within the Luzhou region (2019–2023). In recent years, VTE represents a healthcare concern with potentially grave implications, and the occurrence of acute VTE can lead to fatal outcomes. Hospitalized surgical or medical patients are particularly susceptible to VTE, and this issue persists even after discharge from the

**Table 1. Indications for warfarin prescription in outpatients and inpatients in 2023.**

|  | Outpatient | Inpatient |
|---|---|---|
| Mechanical valve replacement | 4612 | 92 |
| Atrial fibrillation | 1285 | 338 |
| Venous thrombus | 53 | 36 |
| Cerebral infarction | 483 | 161 |
| Postoperative state | 816 | 77 |
| Valvular heart disease | 428 | 399 |
| Others | 75 | 119 |

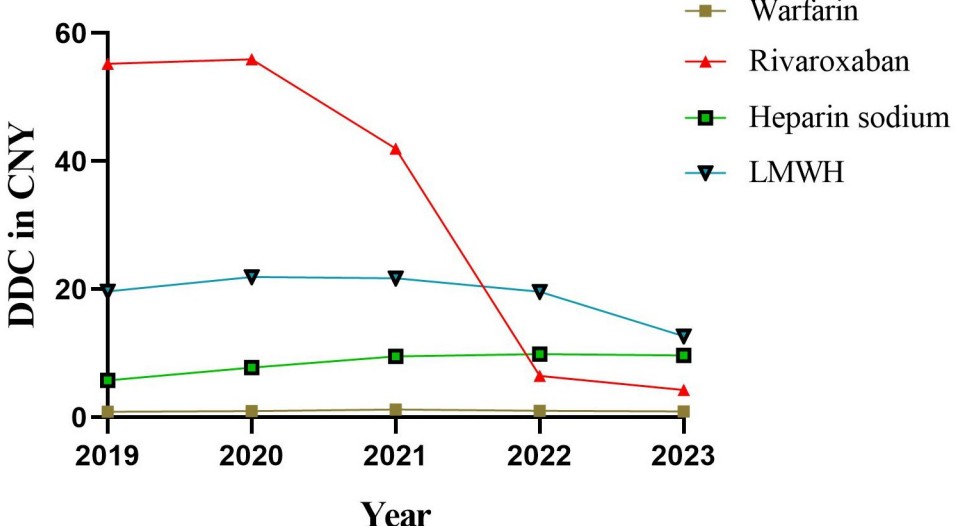

**Fig 6. DDC for four anticoagulants from 2019 to 2023.** DDC: defined daily cost.

hospital [30–32]. In China, the hospitalization crude rate of VTE was 2.9 in 2007 to 15.8 in 2016 per 100,000 population, with a relative increase of five-fold ($p < 0.001$) [33]. Therefore, the utilization of anticoagulants is of utmost importance. The findings from this study demonstrate a notable increase in the utilization of rivaroxaban and LMWH over the past 5 years, accompanied by a decrease in expenditure due to changes in DDC. Simultaneously, their exceptional safety profile and high efficacy distinguish them as superior options among various anticoagulants.

The primary focus of outpatient clinics lies in the management of mild or chronic diseases, wherein patients are typically prescribed self-administered medications. Oral administration is safe and convenient. Hence, in our study, oral anticoagulants such as warfarin and rivaroxaban constitute the primary pharmacological agents for anticoagulation therapy in outpatient settings across eight hospitals. Furthermore, this study revealed a substantial elevation in the outpatient clinic regarding the dosage of LMWH. In a UK national survey and a randomized, controlled trial, outpatient prophylaxis with LMWH demonstrated efficacy in mitigating the risk of VTE among patients with immobilization of the lower extremities [34, 35]. The primary purpose of hospitalization is to address moderate to severe diseases that necessitate systemic treatment, and the treatment period is long. In hospitalized patients, intravenous administration is frequently necessary to promptly alleviate patient symptoms. Furthermore, considering the impaired gastrointestinal drug absorption in certain patients due to underlying diseases, parenteral administration becomes the sole viable option. Consequently, LMWH surpasses other anticoagulants in terms of its utilization as an anticoagulant therapy for hospitalized patients. The selection of different drugs in various environments can effectively demonstrate the role of drug dosage forms and enhance drug efficacy.

The market has become highly competitive with the introduction of DOACs, and warfarin is on the brink of elimination due to its adverse effects and poor compliance [36]. According to the 2021 "Antithrombotic treatment of VTE disease: Second Update of the Chest Guideline and Expert Panel report" [37], DOACs including apixaban, dabigatran etexilate, edoxaban, or rivaroxaban are recommended as superior alternatives to warfarin for initial anticoagulation in patients with VTE. For patients who are contraindicated for DOACs, long-term

administration of warfarin is recommended as an alternative anticoagulation strategy. Therefore, in our study, based on the DDDs of warfarin and rivaroxaban over the past 5 years, it can be observed that the DDDs of warfarin have exhibited a consistent trend, with certain hospitals even experiencing a decline. However, there has been a significant upward trend in the DDDs of rivaroxaban across the eight hospitals since 2021. Furthermore, warfarin exhibits a lightly worse compared to DOACs in terms of adverse effects and clinical efficacy. Patients receiving DOACs exhibited a significantly reduced risk of disease progression and a lower frequency of bleeding events compared to those administered warfarin [38–41]. The adherence to warfarin is suboptimal due to its adverse effects and the necessity for frequent monitoring of INR values. The study demonstrated a significantly higher treatment adherence rate among rivaroxaban users compared to apixaban users (absolute difference (AD): 5.8%; $p < 0.001$), dabigatran users (AD: 9.5%; $p < 0.001$), and warfarin users (AD: 13.6%; $p < 0.001$) [42]. Rivaroxaban does not necessitate routine monitoring, exhibits a lower incidence of adverse drug reactions, and is unencumbered by vitamin K-rich foods, rendering it the preferred anticoagulant over warfarin [43].

Warfarin is the most widely utilized oral anticoagulant globally for the prevention and treatment of various thrombotic events, including deep-vein thrombosis, atrial fibrillation, valvular heart disease and prosthetic valves, as well as peripheral vascular disease [44]. However, according to the expert consensus statement on ICD therapy [45], administration of warfarin for anticoagulation purposes is not recommended following ICD surgery. Therefore, the use of warfarin after ICD implantation in our study was not justified; however, it constitutes a negligible proportion. Despite its flaws, the majority of warfarin diagnoses in this study were in accordance with the prescribed indications for warfarin. However, collaborative efforts are still required to further enhance the rationality of prescribing practices.

Heparin is one of the commonly used anticoagulants for a number of individuals. With the extensive investigation of their pharmacological effects, heparins have also been employed in the treatment of various other diseases. The anticoagulant effect of LMWH is accompanied by its additional properties including anti-inflammatory, antiviral, angiogenesis inhibition, and cell adhesion inhibition [46]. Even the novel coronavirus that emerged in the past few years has a certain effect. Studies have demonstrated that novel coronavirus is capable of inducing pulmonary injury, microvascular endothelial damage, and ultimately thrombosis [47]. The transmission of novel coronavirus is anticipated to result in an upsurge in the demand for heparin medications. Moreover, intravenous access is frequently established in hospitalized patients, and heparin serves as our standard of care for the prevention of venous access thrombosis and the mitigation of catheter-related infections [48]. Therefore, the utilization of heparin used in this study was very large.

The classification of heparin includes unfractionated heparin and LMWH. There are two formulations available for unfractionated heparin, namely heparin sodium and heparin calcium. The utilization of heparin sodium was higher than that of heparin calcium in our study conducted across eight hospitals. According to a study conducted by Lee-White Clotting Time (LWCT), activated partial thromboplastin time (APTT), and thrombin calcium clotting time (TCCT) measurements, subcutaneous injection of both heparin sodium and heparin calcium induced similar anticoagulant responses in normal subjects [49]. However, DDC of heparin sodium (9.54 CNY) was lower than that of heparin calcium (29.12 CNY). In order to alleviate the financial burden on patients, heparin sodium is prescribed more frequently. LMWH because of its less side effects and longer half-life, making it more widely used in hospital and external environment [50, 51]. The pooled analysis involving 1398 postoperative participants demonstrated a significant reduction in the risk of heparin-induced thrombocytopenia with LMWH compared to unfractionated heparin (risk ratio: 0.23, 95%CI: 0.07 to 0.73) [50]. The

DDDs of heparin sodium exhibited an uneven trend across eight hospitals over the past 5 years, whereas LMWH demonstrated a consistent year-on-year increase. These findings suggest that unfractionated heparin is being gradually replaced by LMWH.

In order to ensure equitable accessibility to medications, China has initiated a novel nationwide centralized drug procurement program [52]. The inclusion of drugs in the centralized procurement framework leads to a reduction in their prices, thereby alleviating the financial burden on patients. Therefore, in this study, the DDDs of rivaroxaban and LMWH exhibited an upward trend over the past five years; however, there was a decline in the expenditure and DDC of rivaroxaban and LMWH in 2021 and 2023 respectively. Furthermore, a retrospective study demonstrated that the implementation of a centralized procurement policy resulted in enhancements in adherence to oral anticoagulation therapy and an upsurge in the proportion of patients opting for DOAC [53]. Despite the increasing utilization of DOACs in recent years, warfarin remains a pivotal therapeutic option for patients due to its cost-effectiveness.

The present study also possesses certain limitations. Firstly, the dataset used in this study exclusively comprises tertiary-care hospitals located in Luzhou. Unfortunately, statistics regarding the usage of anticoagulants are not available for secondary hospitals and other hospital levels in Luzhou. However, we contend that tertiary-care hospitals can effectively serve as representative indicators of the medical and drug utilization landscape in Luzhou, thereby allowing for generalizability to other healthcare facilities across China. Secondly, the present study did not encompass the inclusion of antiplatelet medications and thrombolysis, which could potentially impact the analysis pertaining to the utilization of anticoagulants. Thirdly, the retrospective nature of our study, as opposed to a randomized controlled trial, may limit the strength of our findings. Furthermore, our assessment solely focused on the appropriateness of warfarin prescription for its specific indication, without considering the suitability of prescribing alternative anticoagulants. Despite certain limitations, this study offers valuable insights into the contemporary utilization of anticoagulants. In future studies, we will further investigate the adaptability of other anticoagulants in prescription and consider their inclusion in secondary hospitals or continue to explore trends in anticoagulants based on patient demographic characteristics such as age and comorbidities.

## Conclusion

This study shows an increase in anticoagulants utilization and spending from 2019 to 2023. In recent years, the utilization of rivaroxaban and LMWH has been increasing, and they have become the main options in this field. At the same time, the implementation of centralized procurement policy has significantly reduced the price of anticoagulants such as rivaroxaban, thus reducing the economic burden of patients. This study analyzed the development trend of the application of anticoagulants and emphasized the imperative to establish a comprehensive VTE prevention and treatment network centered on primary medical institutions. This approach will effectively guide patients to appropriate medical interventions while ensuring cost containment.

## Supporting information

**S1 Table. Annual trends in the utilization of four anticoagulants at the eight hospitals during the study period.**
(DOCX)

**S2 Table. Inpatient and outpatient DDDs of different anticoagulants in eight hospitals.**
(DOCX)

**S3 Table. Inpatient and outpatient expenditure of different anticoagulants in eight hospitals.**
(DOCX)

## Author Contributions

**Conceptualization:** Hongli Luo.

**Data curation:** Wei Luo, Yan Li, Jiali Yang.

**Investigation:** Yang Liu, Yue Shi.

**Methodology:** Wei Luo, Hongli Luo.

**Visualization:** Wei Luo, Yan Li, Jiali Yang.

**Writing – original draft:** Wei Luo, Yan Li, Jiali Yang.

**Writing – review & editing:** Wei Luo, Hongli Luo.

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
