## [Decision Letter · Decision Letter 0]

14 Nov 2024

PONE-D-24-36242Analysis of anticoagulant drug utilization in eight hospitals within the Luzhou region from 2019 to 2023PLOS ONE

Dear Dr. Luo,

Thank you for submitting your manuscript to PLOS ONE. After careful consideration, we feel that it has merit but does not fully meet PLOS ONE’s publication criteria as it currently stands. Therefore, we invite you to submit a revised version of the manuscript that addresses the points raised during the review process.

We look forward to receiving your revised manuscript.

Kind regards,

Mohammed Hussain Abutaleb, PhD

Academic Editor

PLOS ONE

**Journal Requirements:**

Sichuan Provincial Department of Science and Technology project (No. 2023JDKP0040)

3. In the online submission form, you indicated that The data that support the findings of this study are available from Hospital Information System and Prescription Automatic Screening System of Sichuan Medico Software Research and Development Co., Ltd. but restrictions apply to the availability of these data, which were used under license for the current study, and so are not publicly available. Data are however available from the authors upon reasonable request and with permission of Hospital Information System and Prescription Automatic Screening System of Sichuan Medico Software Research and Development Co., Ltd.

Reviewers' comments:

Reviewer's Responses to Questions

**Comments to the Author**

1. Is the manuscript technically sound, and do the data support the conclusions?

Reviewer #1: Yes

Reviewer #2: Partly

Reviewer #3: Yes

Reviewer #4: Yes

2. Has the statistical analysis been performed appropriately and rigorously? 

Reviewer #1: No

Reviewer #2: I Don't Know

Reviewer #3: Yes

Reviewer #4: N/A

3. Have the authors made all data underlying the findings in their manuscript fully available?

Reviewer #1: Yes

Reviewer #2: No

Reviewer #3: Yes

Reviewer #4: Yes

4. Is the manuscript presented in an intelligible fashion and written in standard English?

Reviewer #1: Yes

Reviewer #2: Yes

Reviewer #3: Yes

Reviewer #4: Yes

5. Review Comments to the Author

**Reviewer #1:** This article provides a valuable and timely analysis of anticoagulant drug utilization trends across multiple tertiary hospitals in the Luzhou region from 2019 to 2023.

However, to strengthen the statistical rigor and enhance the interpretability of the results, I assert necessity to including inferential statistical tests where comparisons are made across groups (e.g., DDD trends, cost differences). Including appropriate inferential tests would allow the authors to draw more robust conclusions regarding observed trends and differences, enhancing the article's scientific validity.

Ethically, the study appears to be conducted with attention to patient privacy, using anonymized data, and the authors have appropriately disclosed the lack of conflicts of interest. I did not observe any issues with dual publication or ethical concerns that would compromise the integrity of the research.

Overall, this study is well-structured, addresses an important healthcare issue, and has the potential to contribute valuable insights to the field of anticoagulation therapy. With the addition of inferential statistics and some clarification in specific sections, this work would represent a strong and impactful addition to the literature.

**Reviewer #2:** This study presented a descriptive analysis of anticoagulants use over 4 years duration in 8 hospitals in Luzhou region in China. The main idea of this study appears clear, but the poor quality of writing could make it challenging for readers to comprehend. I recommend that this study undergo Major editorial and language revisions to enhance clarity and readability.

Find in the attached my comments.

**Reviewer #3: **1. Study aim statement - "This study aims to luzhou area through the analysis of the drug utilization research in tertiary hospital nearly five years of anticoagulant drugs usage, combining economics discussion and thinking of anticoagulant drugs selection and reasonable use." Need to rephrase the aim clearly and precisely.

2. Ethics statement - " The data used in this study were completely anonymized, did not require review and approval

by ethics committees, and the requirement for informed consent was waived" - For retrospective analysis of drug utilization studies, ethical approval may still be needed despite the observational nature of the study. Hence it would be appropriate you declare the proof of waiver status, as confirmed by an ethics board.

3. Figures (ALL) - Please specify the X axis and also Y axis clearly. Also add a brief description on what the figure depicts? (Example: Figure 1: Usage of anticoagulant drugs at eight hospitals and DDDs: defined daily doses)

**Reviewer #4:** Title: The title is concise and captures the manuscript appropriately.

Abstract: This was well-written and sound.

Methods: The methods does not quite capture a particular study design. The authors state that it was a retrospective multicenter study. As to whether it was retrospective cross-sectional or cohort.

Results:This was well written and captures the objectives of the study.

Conclusion:The conclusion is tied in with the initial study objectives.

6. PLOS authors have the option to publish the peer review history of their article (what does this mean?). If published, this will include your full peer review and any attached files.

Reviewer #1: No

Reviewer #2: **Yes: **Alyaa M. Ajabnoor

Reviewer #3: No

Reviewer #4: No

---

## [Author Response · Author response to Decision Letter 0]

7 Dec 2024

Response to Reviewers

Reviewer #1 Comments to the Author: This article provides a valuable and timely analysis of anticoagulant drug utilization trends across multiple tertiary hospitals in the Luzhou region from 2019 to 2023.

However, to strengthen the statistical rigor and enhance the interpretability of the results, I assert necessity to including inferential statistical tests where comparisons are made across groups (e.g., DDD trends, cost differences). Including appropriate inferential tests would allow the authors to draw more robust conclusions regarding observed trends and differences, enhancing the article's scientific validity.

Ethically, the study appears to be conducted with attention to patient privacy, using anonymized data, and the authors have appropriately disclosed the lack of conflicts of interest. I did not observe any issues with dual publication or ethical concerns that would compromise the integrity of the research.

Overall, this study is well-structured, addresses an important healthcare issue, and has the potential to contribute valuable insights to the field of anticoagulation therapy. With the addition of inferential statistics and some clarification in specific sections, this work would represent a strong and impactful addition to the literature.

Reply: Thank you for your valuable scientific comments and endorsement of our manuscript. With your help, we have added inferential statistical tests to enhance statistical rigor and improve the interpretability of the results. Thank you very much for your valuable suggestions on our manuscript.

Comment 1. Line 19 to 20: “and provide valuable insights for their management and rational use”. Please clarify how this study contributes to existing literature on anticoagulant use in hospital settings.

Reply: Thank you for your invaluable advice. We have revised the background of the abstract to explicitly highlight the implications of our article on anticoagulant use in various healthcare settings. The revised background (line:24-27) is as follows:

With the increasing utilization of anticoagulants, the selection of appropriate anticoagulants has emerged as a significant quandary. The objective of this study was to evaluate recent trend in the utilization and expenditure of anticoagulants within a specific region, aiming to provide valuable insights into the optimal choice of anticoagulants across other healthcare facilities.

Comment 2. Line 27: “The DDDC of rivaroxaban decreased”. Adding explanation of this unexpected findings would enlighten reader.

Reply: Thank you for your invaluable advice. We have revised the results to provide a comprehensive explanation of the factors contributing to the decrease in DDC, aiming to enhance readers' comprehension of the article. The revised version (line: 36-41) is as follows:

The implementation of the centralized procurement policy, however, resulted in a decline in the expenditure of rivaroxaban and LMWH in 2021 and 2022 respectively. The DDC value of rivaroxaban experienced a substantial decrease over the past five years (p = 0.020), declining from 55.20 Chinese Yuan (CNY) in 2019 to 4.28 CNY in 2023. Conversely, there was a slight increase noted in the DDC of heparin sodium during this time frame (p = 0.042).

Comment 3. Line 28: “significantly”. Clarify any statistical tests used for comparing DDDC trends across hospitals, as this is not mentioned. Significance cannot be assessed using only descriptive statistics.

Reply: Thank you for your valuable scientific comment. We sincerely apologize for the oversight that led to an error in this section. In order to address this issue, we have added linear regression analysis to the article in order to show the trend of defined daily doses (DDDs) and defined daily cost (DDC), which helps strengthen the scientific rigor of our study. For example (line:242-246):

The DDC value of rivaroxaban exhibited a significant decrease over the past 5 years (p = 0.020), declining from 55.20 CNY in 2019 to 4.28 CNY in 2023. Furthermore, there was a slight increase observed in the DDC of heparin sodium during the same period (p = 0.042), rising from 5.79 CNY in 2019 to 9.67 CNY in 2023. However, the DDC values of warfarin (p = 0.793) and LMWH (p = 0.202) remained relatively stable throughout the past 5 years.

Comment 4. Line 33-36. Conclusion did not reflect on the main results of the study. I think, your conclusion should more directly reflect the study’s key results, highlighting the observed trends in anticoagulant usage, etc. Especially highlighting and explaining the discrepancy found between DDDs and DDDC.

Reply: Thank you for your valuable scientific comment. The conclusions in the abstract have been revised to more accurately reflect the key findings of the study, with the revised conclusions (line: 42-45) presented as follows:

Over the past five years (2019-2023), there has been an increase in the utilization of rivaroxaban and LMWH. However, their expenditure has decreased. In addition, the utilization and expenditure of warfarin and heparin sodium remained relatively stable. The application prospects of rivaroxaban and LMWH are promising.

Comment 5. Line 93-95. It’s helpful to highlight gaps this study fills regarding regional drug utilization. Consider suggesting more details on these gaps for the reader’s understanding. 

Also, please clarify the significance of selecting the Luzhou region and tertiary hospitals for this study.

Reply: Thank you for your valuable advice. We have incorporated the rationales and significance behind the selection of tertiary-care hospitals in Luzhou, and the specific modifications (line:113-121) as follows:

The city of Luzhou, situated in the southern region of Sichuan province, stands as a representative example with relatively high medical and economic standards within the province. Moreover, owing to factors such as advanced medical facilities and convenient transportation, tertiary-care hospitals are predominantly chosen by patients for their treatment needs. Consequently, this inclination towards tertiary-care hospitals results in a greater utilization of anticoagulant medications and facilitates a more comprehensive understanding of their usage trends. Therefore, the objective of this study is to provide an all-encompassing overview regarding the utilization and expenditure patterns of anticoagulant drugs in Luzhou's tertiary-care hospitals from 2019 to 2023 through DUR.

Comment 6. Line 140: “luzhou”. Please Capitalize Luzhou wherever mentioned.

Reply: We agree with you and apologize for our mistake. We have revised it. Thank you for your positive and progressive suggestions on our manuscript.

Comment 7. Line 227: “An estimated 370,000 deaths related to VTE occur annually in six European countries”. Is there any estimation in China?

Reply: Thank you for your valuable advice. We removed the VTE example from Europe and added the epidemiology of VTE in China, with the following changes (line:253-255):

In China, the hospitalization crude rate of VTE was 2.9 in 2007 to 15.8 in 2016 per 100,000 population, with a relative increase of five-fold (p < 0.001)[1].

Reference

[1].Zhang Z, Lei J, Shao X, Dong F, Wang J, Wang D, et al. Trends in Hospitalization and In-Hospital Mortality From VTE, 2007 to 2016, in China. Chest. 2019;155: 342–353. doi:10.1016/j.chest.2018.10.040

Comment 8. Line 228-235. These sentences present general information on VTE prevention and treatment, which does not directly explain or support your study’s findings. Consider moving it to the introduction if you feel it provides necessary background context. In the discussion, focus on interpreting your results in relation to existing research and highlighting specific insights drawn from your data. 

Reply: Thank you very much for your scientific comment. We omitted general information about VTE prevention and treatment and focused on explaining results relevant to existing research and highlighting specific insights. The specific changes are as follows (lines 255-259):

Therefore, the utilization of anticoagulants is of utmost importance. The findings from this study demonstrate a notable increase in the utilization of rivaroxaban and LMWH over the past 5 years, accompanied by a decrease in expenditure due to changes in DDC. Simultaneously, their exceptional safety profile and high efficacy distinguish them as superior options among various anticoagulants.

Comment 9. Line 241-242. What does “nationwide” mean in this context? This is confusing because apparently these two studies were not conducted in China. Please change the sentence to remove that ambiguity. 

Reply: We agree with you, and we apologize for this ambiguous sentence. The specific changes are as follows (line:265-267):

In a UK national survey and a randomized, controlled trial, outpatient prophylaxis with LMWH demonstrated efficacy in mitigating the risk of VTE among patients with immobilization of the lower extremities.

Comment 10. Line 265-267. The term 'AD' is unclear and should be defined for clarity, at least in the first appearance. Additionally, the inclusion of P-values and using the technical term “significant”, suggest inferential statistical analysis, yet your methodology states that only descriptive statistics were used. Please clarify if inferential tests were conducted and, if so, detail these methods in the methodology section. If not, consider removing the P-values and “significant(ly)” terms; to maintain consistency with your stated analysis approach. 

Reply: Thank you very much for your scientific comment. We have redefined 'AD' to make its meaning clearer to the reader. In addition, we added inferential statistical methods and included p values.

Comment 11. Line 270-273. Again, as above. These are general knowledge, and suggested to be moved into background section if necessary. 

Reply: Thank you for your valuable advice. We also agree with you and have deleted most of the content. However, considering that the discussion in this section is mainly to analyze the specific indications for warfarin in order to determine the rationality of its prescription. We also retained some of the necessary content. The specific changes are as follows (line: 295-303):

Warfarin is the most widely utilized oral anticoagulant globally for the prevention and treatment of various thrombotic events, including deep-vein thrombosis, atrial fibrillation, valvular heart disease and prosthetic valves, as well as peripheral vascular disease. However, according to the expert consensus statement on ICD therapy, administration of warfarin for anticoagulation purposes is not recommended following ICD surgery. Therefore, the use of warfarin after ICD implantation in our study was not justified; however, it constitutes a negligible proportion. Despite its flaws, the majority of warfarin diagnoses in this study were in accordance with the prescribed indications for warfarin. However, collaborative efforts are still required to further enhance the rationality of prescribing practices.

Comment 12. Line 308-309. Is there any expected effect of COVID-19 on this trend?

Reply: Reply: Thank you for your valuable scientific advice. Patients with COVID-19 have a certain risk of thrombosis, and the guidelines require that patients with COVID-19 need to use heparin drugs for anticoagulant therapy [2]. In addition, the duration of COVID-19 was from December 2019 to December 2022, which was the window period of our study. Thus, COVID-19 had an effect on trends in the use of heparin.

Reference

[2]. National Health Commission of the People’s Republic of China. Diagnosis and Treatment of COVID-19(Trial Version 10). Chinese Journal of Clinical Infectious Diseases. 2023;16(1):1-9. Doi:10.3760/cma.j.issn.1674-2397.2023.01.001

Comment 13. Line 319-334. Limitations have been addressed very well. However, it may be useful to recommend more specific future research directions, such as including secondary hospitals or exploring anticoagulant trends by patient demographics (age, comorbidities).

Reply: Thank you for your valuable advice and affirmation. We have added more specific future research directions with the following modifications (line:352-354):

In future studies, we will further investigate the adaptability of other anticoagulants in prescription and consider their inclusion in secondary hospitals or continue to explore trends in anticoagulants based on patient demographic characteristics such as age and comorbidities.

Comment 14. Line 339-340:” The implementation of the centralized procurement policy has significantly reduced the prices of anticoagulant drugs”. Good point. Please point to it briefly in the abstract.

Reply: Thank you for your valuable advice and affirmation. We have pointed out in the abstract. Thank you for your positive and progressive suggestions on our manuscript.

Reviewer #2 Comments to the Author: This study presented a descriptive analysis of anticoagulants use over 4 years duration in 8 hospitals in Luzhou region in China. The main idea of this study appears clear, but the poor quality of writing could make it challenging for readers to comprehend. I recommend that this study undergo Major editorial and language revisions to enhance clarity and readability.

Find in the attached my comments.

Reply: Thank you very much for your scientific comment. We apologize for grammatical errors and poor writing quality. With your help, we made significant language changes to the manuscript to improve the readability of the article. Thank you for your positive suggestions on our manuscript.

Comment 1. Line 1 to 2. I advise to change the title to: anticoagulants utilization in eight hospitals within the Luzhou region: descriptive study from 2019 to 2023.

Reply: Thank you for your valuable advice. Integrating the opinions of multiple reviewers, the full text adds inferential statistical analysis on the basis of the first draft, which makes the article more convincing and rigorous. The title of this article is not suitable for descriptive analysis, but we have changed it accordingly (lines: 1-2):

Anticoagulants utilization in eight hospitals within the Luzhou region from 2019 to 2023.

Comment 2. Line 18 to 20: Rephrase the background of the abstract. It’s not supposed to state the objective but rather to highlight the gap in the literature and briefly mention the aim of the study

Reply: Thank you very much for your scientific comment. We modified the background section so that it was no longer a description of the research purpose only, and the revised background was as follows (line:24-27):

With the increasing utilization of anticoagulants, the selection of appropriate anticoagulants has emerged as a significant quandary. The objective of this study was to evaluate recent trend in the utilization and expenditure of anticoagulants within a specific region, aiming to provide valuable insights into the optimal choice of anticoagulants across other healthcare facilities.

Comment 3. Line 22 to 23: “Data on the utilization of anticoagulant drugs in tertiary hospital within a district from 2019 to 2023 were collected” this sentence needs rewording. It’s clearer to say tertiary-care hospitals throughout. 

Reply: Thank you very much for your scientific comment. We modified the sentence to make it more colloquial. The specific changes are as follows (line:28-30):

The data on anticoagulant utilizations in tertiary-care hospitals within a district were collected from January 2019 to December 2023. 

Comment 4. Line 24: “Diagrams illustrating the trend of change were generated”. Change in what? Either clarify the sentence or remove it if it’s not related to the main analysis of this study.

Reply: Thank you for your valuable advice. What we intended to describe was to plot trends in the frequency of use (DDDs), DDC, and cost of anticoagulants. We apologize for any ambiguity. We modified it as follows (lines 31-32):

The trends in the utilization and expenditure of anticoagulants were examined using linear regression analysis.

Comment 5. Line 28: Define CNY on first mention in the abstract.

Reply: We agree with you and have defined "CNY". Thank you for your positive suggestions on our manuscript

---

## [Decision Letter · Decision Letter 1]

16 Jan 2025

Anticoagulants utilization in eight hospitals within the Luzhou region from 2019 to 2023

PONE-D-24-36242R1

Dear Dr. Hongli Luo,

We’re pleased to inform you that your manuscript has been judged scientifically suitable for publication and will be formally accepted for publication once it meets all outstanding technical requirements.

Kind regards,

Mohammed Abutaleb, PhD

Academic Editor

PLOS ONE

Additional Editor Comments (optional):

Reviewers' comments:

Reviewer's Responses to Questions

**Comments to the Author**

1. If the authors have adequately addressed your comments raised in a previous round of review and you feel that this manuscript is now acceptable for publication, you may indicate that here to bypass the “Comments to the Author” section, enter your conflict of interest statement in the “Confidential to Editor” section, and submit your "Accept" recommendation.

Reviewer #1: All comments have been addressed

Reviewer #3: All comments have been addressed

2. Is the manuscript technically sound, and do the data support the conclusions?

Reviewer #1: Yes

Reviewer #3: Yes

3. Has the statistical analysis been performed appropriately and rigorously? 

Reviewer #1: Yes

Reviewer #3: Yes

4. Have the authors made all data underlying the findings in their manuscript fully available?

Reviewer #1: Yes

Reviewer #3: Yes

5. Is the manuscript presented in an intelligible fashion and written in standard English?

Reviewer #1: Yes

Reviewer #3: Yes

6. Review Comments to the Author

Reviewer #1: Authors made a great efforts to address my points. I recommend this manuscript for publication. Thanks for authors.

Reviewer #3: The authors have addressed all issues raised in the previous review. The manuscript may be considered for publication.

7. PLOS authors have the option to publish the peer review history of their article (what does this mean?). If published, this will include your full peer review and any attached files.

Reviewer #1: No

Reviewer #3: **Yes: **Abdul Nazer Ali

---

## [Editor Report · Acceptance letter]

22 Jan 2025

PONE-D-24-36242R1 

PLOS ONE

Dear Dr. Luo, 

I'm pleased to inform you that your manuscript has been deemed suitable for publication in PLOS ONE. Congratulations! Your manuscript is now being handed over to our production team.

Kind regards, 

on behalf of

Dr. Mohammed Abutaleb 

Academic Editor

PLOS ONE